# Sex and APOE Genotype Alter the Basal and Induced Inflammatory States of Primary Microglia from APOE Targeted Replacement Mice

**DOI:** 10.3390/ijms23179829

**Published:** 2022-08-29

**Authors:** Isha Mhatre-Winters, Aseel Eid, Yoonhee Han, Kim Tieu, Jason R. Richardson

**Affiliations:** 1Department of Environmental Health Sciences, Robert Stempel College of Public Health and Social Work, Florida International University, Miami, FL 33199, USA; 2Department of Neurosciences, School of Biomedical Sciences, Kent State University, Kent, OH 44242, USA

**Keywords:** APOE, neuroinflammation, microglia, Alzheimer’s disease, sex, cytokine

## Abstract

The sex and APOE4 genotype are significant risk factors for Alzheimer’s disease (AD); however, the mechanism(s) responsible for this interaction are still a matter of debate. Here, we assess the responses of mixed-sex and sex-specific APOE3 and APOE4 primary microglia (PMG) to lipopolysaccharide and interferon-gamma. In our investigation, inflammatory cytokine profiles were assessed by qPCR and multiplex ELISA assays. Mixed-sex APOE4 PMG exhibited higher basal mRNA expression and secreted levels of TNFa and IL1b. In sex-specific cultures, basal expression and secreted levels of IL1b, TNFa, IL6, and NOS2 were 2–3 fold higher in APOE4 female PMG compared to APOE4 males, with both higher than APOE3 cells. Following an inflammatory stimulus, the expression of pro-inflammatory cytokines and the secreted cytokine level were upregulated in the order E4 female > E4 male > E3 female > E3 male in sex-specific cultures. These data indicate that the APOE4 genotype and female sex together contribute to a greater inflammatory response in PMG isolated from targeted replacement humanized APOE mice. These data are consistent with clinical data and indicate that sex-specific PMG may provide a platform for exploring mechanisms of genotype and sex differences in AD related to neuroinflammation and neurodegeneration.

## 1. Introduction

Alzheimer’s disease (AD) is a multifactorial progressive neurodegenerative disease primarily characterized by two core pathologies—amyloid-β and tau. Evidence over the last decade has suggested that the presence of a sustained immune response is a hallmark pathology in AD accompanied by activation of microglia, the resident immune cells of the brain [1]. Whole-genome sequencing and genome-wide association studies (GWAS) have identified more than 20 gene variants associated with AD [2,3]. Several of these genes, including *TREM2*, *CR1*, *MS4A*, *BIN1*, and *CD33*, are involved in regulating immune and inflammatory responses, and some are directly associated with the microglial cell type in the brain [4,5,6,7,8]. Integrative systems-based analyses of gene regulatory networks in post-mortem human late-onset AD (LOAD) brain specimens have identified immune-microglial molecular networks as most highly associated with pathophysiology in LOAD [9]. Identification of these genes and their association with AD strengthens the need to investigate the role and mechanism of the innate immune system in the pathophysiology of AD [10,11].

Apolipoprotein E (APOE) is a polymorphic gene in humans coding for three common isoforms, APOE2 (E2), APOE3 (E3), and APOE4 (E4), which vary at positions 112 and 158 for their allelic frequency in the population in the order E3> E4> E2 [12,13,14]. APOE3 is the most common allele, occurring in about 78% of the general population and 59% of the AD population [15]. Due to its high allelic frequency and lack of a strong association with diseases, APOE3 is considered the “wild-type” isoform in humans [16]. APOE4 is the ancestral isoform [17] present in approximately 14% of the general population and 37% of the AD population [15,18,19]. In contrast, APOE2, present in 9% of the general population and 5% of all AD cases, has been documented to be protective for AD [20]. APOE4 has emerged as the strongest genetic risk factor for LOAD over the years through numerous clinical, pathological, epidemiological, and GWAS analyses [8,21]. A single inherited copy of the APOE ε4 allele increases the risk of developing LOAD by ~3–4-fold, while two inherited copies increase the risk of the disease by 8–12-fold [22,23,24].

Although APOE is primarily involved in lipid and cholesterol homeostasis, recent evidence from AD patients and mouse models of AD has identified genotype-specific effects on the AD pathogenesis, including inflammation [25,26,27,28,29,30]. While APOE is expressed in neuronal cells following injury [31] and is predominantly expressed in and secreted by astrocytes [32,33], the presence of APOE has also been reported in microglia [34]. The APOE4 genotype has been demonstrated to modulate neuroinflammation via microglial pathways in human AD patients as well as AD mouse models [35,36]. In mouse primary microglia, inflammatory processes induced by LPS alone or in combination with IFNg increased the secretion of pro-inflammatory cytokines and nitric oxide (NO) production in an APOE genotype-dependent manner, with APOE4 promoting the strongest pro-inflammatory response [37,38,39]. In the CNS, microglia-mediated neuroinflammatory responses are differentially modulated by the APOE genotype, and APOE4 is associated with increased microgliosis and neuroinflammation [35].

Females are at a greater risk of developing AD. According to the 2022 Alzheimer’s Association report, a staggering 3.5 million women (over 2/3 of patients) live with AD [13]. Studies have indicated that women at 45 years of age have a 19.5% lifetime risk of developing AD compared to 10.3% in males, and these numbers increase with age to 21.1% and 11.6%, respectively [13,40]. Reports from longitudinal neuroimaging studies on an Alzheimer’s disease neuroimaging initiative (ADNI) cohort indicate that females with AD accumulate higher levels of pathological proteins (Aβ and tau), have 1–1.5% faster atrophy rates in the hippocampus than males, and have worse cognitive abilities [41,42]. Accumulation of Aβ and tau are more pronounced in female APOE4 carriers than their male counterparts, suggesting a sex-by-genotype interaction of APOE4 in females [41,43,44,45]. Since the immune system plays a crucial role in the progression of AD, it is plausible to hypothesize that females have a greater predisposition to neuroinflammation in AD. Evidence from aging studies as well as AD models suggests that microglia play a key role in modulating inflammatory responses in females. Upregulation of microglia-specific immune genes, including *Aif1* (Allograft inflammatory factor 1/Iba1) and *Tyrobp* (tyrosine kinase binding protein/DAP12), has been reported in older females compared to age-matched males [46]. Sexual divergence has also been observed in AD patients and AD mouse models. Female microglia are more activated and differentially express microglial markers, change in density and morphology with thicker and longer processes, and demonstrate a switch in function from homeostatic to phagocytic [47,48,49].

The humanized targeted APOE mouse model (TR-APOE) was developed by introducing the human APOE gene to replace the murine APOE gene to study the APOE genotype effects on atherosclerosis [50]. Although these mice lack the aggressive genetic mutations that drive early-onset AD, APOE genotype-specific effects have been observed in AD pathogenesis, including synaptic dysfunction [51], cholesterol [52], and lipid metabolism [53], with some studies highlighting interaction between the APOE genotype, age, and sex [54,55]. Differential effects of the APOE genotype on inflammation have also been explored in this model. TR-APOE mice that were chronically administered lipopolysaccharide (LPS) showed increased levels of pro-inflammatory cytokines TNFa and IL6 in both the serum and brain [56]. Microglia derived from mixed-sex TR-APOE4 mice displayed greater secretion of nitric oxide compared to the APOE3 genotype [57]. Unfortunately, previous studies investigating APOE genotype-driven inflammatory differences in TR-APOE microglia have not assessed sex-specific variations [37,57,58]. This is particularly important because females are at higher risk of developing AD [59,60], the female sex significantly affects the APOE4 risk of AD [44,61], and sex has been indicated to modulate the microglial phenotype [62,63].

Here, we sought to determine the sex- and genotype-specific effects of LPS or LPS + interferon-gamma (IFNg) using traditional mixed-sex and sex-specific APOE3 and APOE4 primary microglia (PMG) isolated from TR-APOE3 and -APOE4 mice. Our findings reveal differences in the inflammatory profiles of the genotypes influenced by sex, a key variable not considered in previous studies. Furthermore, our results indicate that LPS or LPS + IFNg induces inflammation that is affected by both the APOE genotype and sex, with a greater inflammatory profile observed with the APOE4 genotype and female sex in primary microglia. Collectively, these data demonstrate that multiple factors contribute to susceptibility to neuroinflammation and they provide an insight into the role of the sex and genotype in affecting this susceptibility.

## 2. Results

### 2.1. APOE4 Genotype Has an Increased Inflammatory Response to LPS in Mixed-Sex Microglia

To investigate the effect of the APOE genotype on inflammation, microglia (PMG) from whole neonatal mouse brains were cultured (Figure 1).

We first examined basal inflammatory gene expression levels in non-stimulated APOE3 and APOE4 microglia (Figure 2).

Analyses of basal gene expression were carried out on dCT values normalized to *Rpl13a* and compared using a two-tailed unpaired Student’s t-test. Basal expression of pro-inflammatory factors increased by 2-fold for *Tnfa* (*p* < 0.0001), 1.5-fold for *Il1b* (*p* = 0.0002), 1.8-fold for *Nos2* (*p* = 0.0005), and 2.4-fold for *Mcp1* (*p* < 0.0001) in the APOE4 microglia compared to APOE3. Similarly, basal differences in the anti-inflammatory cytokines were assessed, and significant increases in the baseline expression of *Arg1* (*p* < 0.0001), *Mrc1* (*p* < 0.0001), *Igf1* (*p* < 0.0001), and *Ym1* (*p* = 0.0002) were observed with 2.2, 1.5, 1.6, and 1.7-fold higher expression, respectively, in the APOE4 microglia. These basal differences suggest that APOE4 PMG have innately increased baseline levels of pro- and anti-inflammatory cytokine expression. Although these increases are moderate, they may contribute to increased activation of PMG following an inflammatory stimulus.

After treatment of mixed-sex PMG with LPS (10 ng/mL), alone or in combination with IFNg (10 mg/mL), for 6 h, an increase in the mRNA expression of pro-inflammatory cytokines was observed in both APOE3 and APOE4 genotypes compared to their respective controls (Figure 3A).

Two-way analysis of variance (ANOVA) indicated a significant genotype effect for pro-inflammatory *Il6* (F_1,24_ = 43.12, *p* < 0.0001), *Il1b* (F_1,24_ = 154.4, *p* < 0.0001), *Tnfa* (F_1,24_ = 153, *p* < 0.0001), *Nos2* (F_1,24_ = 13.51, *p* = 0.0012), *Ifng* (F_1,24_ = 51.96, *p* < 0.0001), and *Mcp1* (F_1,24_ = 46.47, *p* < 0.0001) mRNA levels, with a higher increase observed in the APOE4 microglia. After exposure to LPS alone, APOE4 microglia showed higher mRNA levels of pro-inflammatory cytokines compared to APOE3 microglia. Expression increased by approximately 1.3-fold in *Mcp1* (*p* = 0.002), 1.2-fold in *Nos2* (*p* = 0.05), 1.3-fold in *Ifng* (*p* = 0.04), 1.5-fold in *Il6* (*p* < 0.0001), 1.7-fold in *Il1b* (*p* < 0.0001), and 3-fold in *Tnfa* (*p* < 0.0001) compared to APOE3 microglia treated with LPS. Similarly, after a combination of LPS and IFNg treatment, APOE4 microglia showed an approximately 1.3-fold increase in *Mcp1* (*p* = 0.0001), a 1.2-fold increase in *Nos2* (*p* = 0.0085) and *Il6* (*p* = 0.0002), a 1.5-fold increase in *Ifng* (*p* < 0.0001), and a 1.6-fold increase in *Il1b* (*p* < 0.0001) and *Tnfa* (*p* <0.0001).

For anti-inflammatory cytokines analyzed after exposure to LPS or LPS + IFNg, a significant genotype effect was observed for *Il10* (F_1,23_ = 765.5, *p* < 0.0001), *Arg1* (F_1,24_ = 769.5, *p* < 0.0001), *Mrc1* (F_1,24_ = 190.8, *p* < 0.0001), *Ym1* (F_1,24_ = 1138, *p* < 0.0001), and *Igf1* (F_1,24_ = 582.6, *p* < 0.0001) (Figure 3B). After exposure to LPS alone, a 3.2-fold increase in *Il10* (*p* < 0.0001) was observed in the APOE4 microglia compared to the APOE3 genotype. Smaller, yet significant, increases ranging from 1.5- to 2-fold were seen in APOE4 microglia in *Igf1* (*p* < 0.0001), *Arg1* (*p* < 0.0001), *Mrc1* (*p* = 0.004), and *Ym1* (*p* < 0.0001) compared to APOE3 microglia. However, a significant decrease in mRNA levels of *Fizz1* (*p* = 0.04) was observed in the APOE4 genotype after LPS treatment, whereas no statistical difference was observed for *Tgfb* (*p* = 0.07). In combination with IFNg, LPS increased the expression of *Il10* by 2.5-fold (*p* < 0.0001) in APOE4 microglia. Additionally, *Arg1* (*p* < 0.0001) and *Mrc1* (*p* = 0.0002) expression increased 1.5-fold, whereas *Ym1* (*p* < 0.0001) and *Igf1* (*p* < 0.0001) mRNA levels increased 2-fold in APOE4 microglia. Similar to the LPS treatment, expression of *Fizz1* decreased (*p* = 0.0001), and no change was observed in *Tgfb* (*p* = 0.22). These data are summarized as a heat map in Figure 3C. Together, these data suggest that the microglial response to inflammation is influenced, at least in part, by the APOE genotype.

### 2.2. APOE4 Genotype Elevates Secretion of Inflammatory Cytokine in Mixed-Sex Microglial Cultures

To explore APOE genotype-specific production of inflammatory mediators, microglia were stimulated with LPS for 24 h in the presence or absence of IFNg, and the media were then collected. Media nitrite levels were significantly higher in the APOE4 microglia compared to the APOE3 microglia when we conducted two-way ANOVA (F_1,32_ = 61.85, *p* < 0.0001) following LPS treatment alone (*p* < 0.0001) or in combination with IFNg (*p* < 0.0001) (Figure 4).

In addition to NO, microglia secrete inflammatory factors as a response to an inflammatory stimulus. To assess whether the APOE4 microglia released higher levels of cytokines, media collected after treatments were analyzed for both pro-inflammatory (IL6, TNFa, IL1b, IFNg, MCP1, and MIP1a) and anti-inflammatory (IL10 and IL4) cytokines on the MSD platform. Basal secretion of IL1b, IL6, IFNg, TNFa, MCP1, MIP1a, IL10, and IL4 was higher in APOE4 microglia by 3.5, 2.6, 3.2, 3.3, 3.9, 1.3, 1.7, and 3.5-fold, respectively (Figure 4B). An increase in the secretion of pro- and anti-inflammatory cytokines was detected after treatment with LPS or LPS + IFNg in both APOE3 and APOE4 microglia (Figure 4C). However, these treatments induced significantly higher secretion of these cytokines from APOE4 microglia compared to APOE3. Treating APOE4 microglia with LPS, increased pro-inflammatory IL1b by 1.5-fold (*p* = 0.02), IL6 by 2.3-fold (*p* < 0.0001), TNFa by 4-fold (*p* < 0.0001), IFNg by 1.9-fold (*p* < 0.0001), MCP1 by 1.2-fold (*p* < 0.0001), and MIP1a by 2-fold (*p* < 0.0001), and anti-inflammatory IL10 and IL4 by 3-fold (*p* < 0.0001) and 1.7-fold (*p* < 0.0001), respectively. Additionally, a combination of LPS and IFNg increased pro-inflammatory IL1b by 2.6-fold (*p* < 0.0001), IL6 by 1.4-fold (*p* < 0.0001), TNFa by 2.4-fold (*p* < 0.0001), IFNg by 1.5-fold (*p* < 0.0001), and MCP1 by 1.2-fold (*p* < 0.0001), along with anti-inflammatory IL4 by 1.5-fold (*p* < 0.0001). While there was a 1.6-fold increase in MIP1a and a 1.3-fold increase in IL-10 following treatment with LPS + IFNg, these data were not statistically significant by two-way ANOVA.

### 2.3. Basal Inflammatory Gene Expression Is Driven by APOE Genotype and Sex

The impact of sex on the AD pathology and inflammation is an important factor since the trajectories for progression of the pathology in males and females are distinct. To evaluate the contribution of the APOE genotype and sex in PMG to an inflammatory stimulus, sex-specific primary cultures were generated from TR-APOE3 and APOE4 mouse pups. The percentage of purity of microglia in cultures was assessed (Figure 1C,D), and cells were validated for sex-specificity by examining the presence or absence of the sex-determining region-Y (*Sry*) gene by qPCR (Figure 1E). Gene expression in non-stimulated male and female APOE3 and APOE4 microglia was measured after 6 h to evaluate differences in the basal responses between genotypes and sexes (Figure 5).

Results were normalized to the APOE3 male group, and basal mRNA levels were compared by two-way ANOVA with Tukey’s post-hoc testing. Differential responses in sex-specific microglia from APOE4 mice were observed with an increase in female *Il1b, Tnfa, Il6,* and *Nos2* by 2–3-fold, and *Mcp1* by 1.7-fold, compared to male microglia (Figure 5A). Furthermore, a significant interaction between the APOE genotype and sex was detected for *Il1b* (F_1,12_ = 144.9, *p* < 0.0001), *Tnfa* (F_1,12_ = 532, *p* < 0.0001), *Il6* (F_1,12_ = 141.8, *p* < 0.0001), *Nos2* (F_1,12_ = 135, *p* < 0.0001), and *Mcp1* (F_1,12_ = 40.44, *p* < 0.0001). Female microglia derived from APOE4 mice showed a higher gene expression in *Il1b* by 5.2-fold, in *Tnfa* and *Nos2* by 3-fold, in *Il6* by 3.8-fold, and in *Mcp1* by 2-fold compared to APOE3 females. This suggests that in addition to the APOE genotype, sex also plays a key role in modulating inflammation.

The basal gene expression profiles for homeostatic and anti-inflammatory genes were also investigated in APOE3 and APOE4 male and female microglial cultures (Figure 5B). Significant interaction between the genotype and sex was detected for *Il10* (F_1,12_ = 9.54, *p* = 0.009), *Mrc1* (F_1,12_ = 93.25, *p* < 0.0001), *Igf1* (F_1,12_ = 87.72, *p* < 0.0001), *Tgfb* (F_1,12_ = 45.01, *p* < 0.0001), and *Fizz1* (F_1,12_ = 13.92, *p* = 0.0029). Basal gene expression for female APOE4 was upregulated compared to male APOE4 for *Il10* (*p* = 0.006), *Arg1* (*p* = 0.006), *Mrc1* (*p* < 0.0001), and *Tgfb* (*p* = 0.0015), and no change was observed in *Fizz1, Igf1*, or *Ym1*. Contrastingly, gene expression in APOE3 females was downregulated for *Mrc1* (*p* = 0.0055), *Igf1* (*p* < 0.0001), and *Tgfb* (*p* = 0.004) compared to APOE3 males, whereas no difference was observed for *Il10* or *Ym1*. These basal differences between the sexes of each genotype indicate that distinct responses to inflammation may be observed in part due to varying basal levels of homeostatic and anti-inflammatory gene expression.

### 2.4. Sex Differences in Inflammatory Gene Expression of Immune-Stimulated TR-APOE3 and TR-APOE4 Microglia following Stimulus

Subsequently, the differences in inflammatory gene expression, induced via LPS alone or a combination of LPS and IFNg, were investigated and analyzed using three-way ANOVA with Tukey’s post-hoc testing. Pro-inflammatory cytokine mRNA expression in microglia was detected in the order E4 female > E4 male > E3 female > E3 male (Figure 6A). Significant interactions between treatment and sex, treatment and genotype, as well as treatment, genotype, and sex were observed for most cytokines. Following LPS treatment, sex-specific differences were observed in both APOE3 and APOE4 microglia. Pro-inflammatory cytokines increased 1 to 1.5-fold in APOE3 female microglia compared to male microglia, with a similar increase observed in APOE4 microglia. Comparison of female cultures of APOE3 and APOE4 microglia revealed a 1.5- to 2-fold increase in APOE4 female pro-inflammatory mRNA levels. A similar pattern of fold increase was observed after treatment with LPS + IFNg (Figure 6A).

Following treatment with LPS or LPS + IFNg, gene expression of homeostatic and anti-inflammatory genes was upregulated in the order E4 female > E4 male > E3 female > E3 male (Figure 6B). A significant interaction between treatment and sex, treatment and genotype, as well as treatment, genotype, and sex, was observed for the majority of cytokines. LPS treatment induced a 1.5- to 2-fold increase in female microglia cultures of each genotype compared to the male cultures of the corresponding genotype. Comparison between the female cultures of both genotypes indicated a 2-fold increase in *Il10* (*p* < 0.0001), *Igf1* (*p* < 0.0001), *Ym1* (*p* = 0.02), and *Tgfb* (*p* < 0.0001), and a 3-fold increase in *Arg1* (*p* < 0.0001) in the APOE4 female microglia. Cultures treated with LPS + IFNg showed similar results. These data are summarized as a heat map in Figure 6C. Together, these data indicate that several pro-inflammatory and anti-inflammatory genes are upregulated in female APOE4 microglia, and exposure to an inflammatory stimulus exacerbates the induction of these genes.

### 2.5. Female APOE4 Microglial Cultures Have Increased Secretion of Inflammatory Cytokines

Nitric oxide levels were measured indirectly by quantifying the levels of media nitrite in media collected from APOE3 and APOE4, male and female microglial cultures that were >98% pure. Data analysis using three-way ANOVA indicated a treatment effect dependent on the genotype (F_3,48_ = 37.83, *p* < 0.0001) and sex (F_3,48_ = 12.52, *p* < 0.0001). LPS treatment alone increased media nitrite levels significantly in APOE4 male and female microglial cultures compared to the respective vehicle control, whereas LPS + IFNg increased media nitrite in both APOE3 and APOE4 male and female cultures (Figure 7A). Furthermore, female APOE4 microglia secreted 2.6-fold more media nitrite compared to APOE3 females and 1.7-fold more compared to APOE4 male cultures.

Accumulation of other inflammatory factors was also evaluated in the male and female cultures using MSD multiplex assays (Figure 7B,C). Basal secretion of IL1b, IL6, TNFa, MCP1, MIP1a, and IL10 was higher in APOE4 microglia (Figure 7B). Yet, female cultures from both genotypes secreted approximately 1.5 to 2-fold higher levels of these cytokines, indicating sexual dimorphism in response to inflammation following treatment with LPS or LPS + IFNg. A treatment by genotype by sex effect was observed for pro-inflammatory cytokines including IL6 (F_2,60_ = 48.59, *p* < 0.0001), TNFa (F_2,60_ = 163, *p* < 0.0001), MCP1 (F_2,60_ = 11.97, *p* < 0.0001), and MIP1a (F_2,60_ = 19.36, *p* < 0.0001), as well as anti-inflammatory IL10 (F_2,84_ = 25.56, *p* < 0.0001) and IL4 (F_2,36_ = 9.87, *p* = 0.0004). APOE4 female microglia had the highest accumulation of both pro- and anti-inflammatory cytokines compared to APOE3 male and female microglia, as well as APOE4 male microglia (Figure 7C).

### 2.6. APOE Genotype- and Sex-Dependent Inflammation Is Associated with NF-κB p65 Levels

LPS is recognized by toll-like receptor 4 (TLR4), which triggers signaling cascades, leading to nuclear translocation of NF-κB and transcriptional induction of cytokines and chemokines (canonical pathway) [64]. Nuclear factor-κB (NF-κB) is an inducible transcription factor that regulates pro-inflammatory genes and plays a vital role in the activation of immune cells, including microglia [64]. It comprises five subunits, including p65, which is activated in response to stimuli such as LPS [65]. We, therefore, sought to determine whether the sex-specific inflammatory profile of APOE3 and APOE4 microglia was mediated through p65 activation.

Immunofluorescent staining demonstrated an increase in the fluorescence intensity of total p65 basally (control) in the APOE4 cultures, with greater levels in APOE4 female PMG compared to APOE4 male cultures (Figure 8A,B). Three-way ANOVA indicated a treatment effect on the genotype (F_1,16_ = 19.31, *p* = 0.0005), with greater levels of total p65 in LPS + IFNg treated APOE4 male and female PMG compared to APOE3 PMG. Quantification of total p65 showed that LPS + IFNg increased p65 in both male and female cultures of APOE3 (male: *p* = 0.004; female: *p* < 0.0001) and APOE4 (male: *p* < 0.0001; female: *p* < 0.0001) compared to the respective vehicle control (Figure 8B). Sex differences in p65 levels were also observed in APOE4 PMG, where female microglia treated with LPS + IFNg showed 1.3-fold greater p65 levels than males (*p* < 0.0001). No differences were observed between APOE3 male and female PMG.

Since activation of NF-κB involves nuclear translocation of the p65 subunit, we assessed nuclear p65 levels by quantifying the fluorescent intensity of p65 in the DAPI channel. Similar to total p65, higher levels of nuclear p65 were observed basally in the APOE4 females compared to APOE4 males (Figure 8C). Treatment with LPS + IFNg increased the translocation of p65 into the nuclei of both male and female APOE3 (male: *p* = 0.04; female: *p* = 0.0006) and APOE4 (male: *p* = 0.0006; female: *p* < 0.0001) PMG compared to their respective controls. Three-way ANOVA indicated a genotype-by-sex effect (F_1,16_ = 4.83, *p* = 0.043), with approximately 1.3-fold greater nuclear p65 levels observed in female APOE4 PMG compared to male APOE4 cultures (*p* = 0.0243). No statistically significant difference was noted between the APOE3 male and female cultures.

## 3. Discussion

Meta-analysis of several GWAS in AD has indicated APOE4 is the most significant genetic risk factor across populations, influencing the prevalence and age-at-onset in an isoform-dependent manner [21,22,66]. Evidence from studies across neurodegenerative and neuropathological diseases indicates that APOE4 is also a risk factor in cerebral amyloid angiopathy (CAA), Lewy body dementia, vascular dementia, and tauopathy [67]. Additionally, APOE4 is associated with an increased risk of cardiovascular diseases and type 2 diabetes [68]. Contrastingly, in age-related macular degeneration (AMD), evidence suggests that APOE4 is protective [67]. Interaction between race/ethnicity and the effect of the APOE genotype on AD risk has previously been studied and reviewed [69]. Evidence from clinical and autopsy-based studies indicated that in white individuals, the risk of AD was increased with E4 allele dosage, from one copy (E2/E4, OR 2.6; E3/E4, OR 3.2) to two copies (E4/E4, OR 14.9), compared to the E3/E3 genotype [61]. Conversely, the E2 allele lowers the risk of developing AD (E2/E2 and E2/E3, OR 0.6) compared to the E3/E3 genotype [61]. Population-based studies have demonstrated that the APOE4-AD association is weaker in African American (E4/E4, OR 5.7) and Hispanic populations (E4/E4, OR 2.2), whereas in the Japanese E4 carriers, the risk was found to be highest (E4/E4, OR 33.1) [61,70,71]. Recent evidence from a longitudinal cohort study of a Hispanic/Latino population indicated that APOE4 was associated with an increased risk of cognitive decline (OR 1.15), with the strongest association in Cubans (OR 1.46) [72]. Contradicting this, APOE4 has been shown to have a protective effect in a non-industrialized Tsimane population of the Bolivian Amazon with a high parasitic burden [73].

Recent GWAS have identified more than 20 gene variants associated with AD, of which several genes, including TREM2, CD33, and CLU, are involved in regulating immune responses [2] and are directly associated with microglia [7]. Neuroinflammation is a concerted process in which microglia play a key role in regulating the immune response. Activation of the innate immune system, reflected by a milieu of pro- and anti-inflammatory cytokines and chemokines, is a hallmark pathological feature of several neurodegenerative diseases, including AD [74]. The transcriptional profile of microglia in AD shifts from a homeostatic phenotype to inflammatory modules such as disease-associated microglia (DAM) [75,76,77], human Alzheimer’s microglia (HAM) [78], or microglial neurodegenerative phenotype (MGnD) [36]. Interestingly, APOE has been shown to regulate some of these microglial transcriptional signatures, as demonstrated in postmortem human brain studies, AD mouse models, and in vitro microglial cultures derived from both humans and mice [36,75,79,80].

In this study, we first evaluated the inflammatory response of mixed-sex microglia generated from both TR-APOE3 and APOE4 mice. Consistent with prior literature, we observed an increase in the inflammatory profile of APOE4 microglia both basally and following stimulation with LPS or a combination of LPS and IFNg. While APOE4 microglia have been implicated in exacerbating systemic as well as neuroinflammation, the genotype-dependent effects on neuroinflammation have only been studied in a mixed-sex microglial culture. To address this, we next investigated the differential inflammatory response of microglia based on the APOE genotype and sex of TR-APOE mice. Across both sexes and genotypes, we observed an increase in inflammation post-LPS treatment. This upregulation was found to be sex-specific for both genotypes, with females exhibiting greater inflammatory gene expression and cytokine secretion. Lastly, we report that sex and genotype together influence the microglial inflammatory profile, with APOE4 female microglia showing elevated inflammatory cytokines and gene expression compared to APOE4 male and APOE3 male and female microglia.

The APOE4 genotype has been associated with enhanced immune responses in humans, mouse models, and in vitro cultures [79,81,82]. Glial cultures consisting of both microglia and astrocytes generated from Sprague–Dawley rats produced higher levels of IL1b after exposure to recombinant APOE4 [81]. Although statistical significance was not reached, enhanced production of several cytokines and chemokines was seen in human whole blood of APOE4 carriers [82]. Human clinical data from AD patients reported elevated plasma levels of the pro-inflammatory cytokines TNFa, IL6, and IL1b in APOE4 carriers, carrying either one or two copies, compared to APOE2 and APOE3 carriers [83,84]. In agreement with this, we observed basal elevation in the media levels and gene expression of pro-inflammatory cytokines, including in the non-stimulated APOE4 microglia, compared to the APOE3 genotype. This suggests that APOE4 microglia have an inherently increased pro-inflammatory phenotype. A recent study by Lanfranco and colleagues in TR-APOE mice showed no differences in basal TNFa mRNA across genotypes [58]. In contrast, we observed a 1.5- to 2-fold increase in the basal TNFa mRNA of APOE4 microglia compared to APOE3. A possible reason that could largely contribute to conflicting findings is the variability in the method used to generate primary microglia (CD11b targeted magnetic separation yielding >98% purity, or the flask shaking method yields a 90–95% enriched microglial culture). Previous studies that evaluated the effect of inflammation on microglia in an APOE isoform-dependent manner saw similar genotype effects with high concentrations of LPS (100–500 ng/mL) [38,58], although concentrations as low as 5 ng/mL have induced substantial inflammation [58]. Furthermore, on investigating the cultures from both genotypes in a sex-specific manner, we found that both APOE3 and APOE4 females had a more pronounced inflammatory profile. Importantly, this is consistent with clinical data from healthy individuals carrying the APOE3 or APOE4 gene; APOE4 females show a greater increase in both pro-inflammatory cytokine release and gene expression [82,85,86].

Cultured mixed-sex microglia from TR-APOE mice have been reported to induce media NO following treatment with IFNg or IFNg + polyinosinic: polycytidylic acid (PIC, a TLR3 ligand), with higher levels observed in APOE4 cultures [57]. Similarly, IFNg alone or in combination with LPS increased media nitrite and NOS2 mRNA levels in APOE4 PMG [37]. These effects were observed at a high (100 ng/mL) dose of LPS or at a later timepoint [37,38]. Our findings from mixed-sex cultures treated with LPS or LPS + IFNg are consistent with the previously published reports, and we indeed observe increased NO levels and an approximately 1.2-fold increase in NOS2 mRNA with APOE4 microglial cultures. Additionally, we also note a basal increase in the NOS2 mRNA levels of APOE4 microglia. An increase in the pro-inflammatory cytokines, such as TNFa and IL6, has additionally been reported in the supernatant of APOE4 microglia [37]. We validated these results in the media from mixed-sex APOE4 cultures, and we noted an increase in the gene expression of these cytokines.

Furthermore, we identified novel sex differences in the response of microglia to inflammation. The observed sex differences are apparent in both APOE3, and APOE4 genotypes, with the greatest induction of inflammation observed in APOE4 female microglia. While the specific mechanism elucidating the interaction between the APOE4 genotype and female sex remains ambiguous, our findings are consistent with largely recognized studies in humans and TR-APOE mice indicating sex and APOE genotype differences in the inflammatory pathways [87,88,89]. Sex-based differences in the microglial response to inflammation have been reported, with studies indicating that females are predisposed to a more robust inflammatory response than males [46,63,90]. In addition, transcriptomic analyses in adult microglia have revealed significant sex differences that are at least partially independent of adult sex hormone exposure [46,62]. In this investigation, to reduce the effects of exogenous hormones on cultures, we supplemented the growth and treatment with charcoal-stripped FBS, suggesting that the effects we observed are predominantly the result of intrinsic sex differences.

Microglia have been implicated as potential modulators of sex differences and inflammatory profiles in neurodegenerative diseases, including AD [36,63]. Abundant evidence from previous human studies indicates sex differences stratified by the APOE genotype in AD [41]. While much of the female predisposition to AD has been reflected in longevity, mounting evidence supports the notion that other factors, including inflammation, may also potentiate the risk of AD for females [49]. Assessment of microglial metabolism across AD patients and mouse models has revealed that female microglia are more activated than those of males and present increased inflammatory gene expression [48]. We speculate that these sex-specific responses in AD likely involve, but are not limited to, the role of microglia.

Although the mechanism through which APOE isoforms differentially modulate neuroinflammation is unclear, the APOE protein has been demonstrated to exert immunomodulatory effects [56]. APOE has been shown to reduce microglial activation and subsequent release of inflammatory mediators in an APOE genotype-specific manner [91,92,93]. Recently, Lanfranco and colleagues showed that microglia derived from TR-APOE4 mice had significantly lower levels of secreted and cellular APOE compared to the APOE2 genotype [58]. While no difference was observed between the APOE3 and APOE4 genotypes, using highly purified microglial cultures (>99%) might provide additional insights into potential differences. The researchers further found that stimulating the microglia with LPS increased secretion of APOE in the APOE2 and APOE3 microglia, with no changes observed in the APOE4 microglia. Conversely, exposure to TNFa increased APOE secretion in the APOE4 microglia, but secretion from APOE2 and APOE3 remained unchanged. Overall, these data indicate isoform-dependent regulation of APOE in microglia, which is further modified by different inflammatory insults that may exacerbate neuroinflammation. Studies assessing APOE levels in brain lysates of TR-APOE mice indicated reduced levels of the APOE protein in APOE4 mice compared to the APOE3 genotype [37]. However, even after controlling for APOE levels, higher pro-inflammatory responses were observed in the APOE4 mice, suggesting that APOE4 may regulate the immune response independent of the APOE protein concentration [37].

Another potential mechanism by which APOE4 may result in an enhanced response to inflammatory stimuli is through various signaling pathways. LPS activates CD14/TLR4 coreceptors, leading to increased inflammation in either a p38MAPK- or NF-κB-dependent manner [38]. Maezawa and colleagues demonstrated that the APOE4 microglia differentially regulate the p38 mitogen-activated protein kinase signaling pathway, subsequently resulting in increased cytokine secretion by APOE4 microglia compared to APOE3 microglia [38]. A growing body of evidence has also shown that APOE4 enhances inflammation in the brain by modulating the signaling of NF-κB, which is a crucial transcription factor regulating immune gene expression [94,95]. Importantly, these mechanisms have not been previously explored in a sex-specific manner. In this study, we semi-quantified p65 levels of NF-κB by immunocytochemistry to identify its role in inducing the APOE genotype and sex-dependent inflammation in primary microglia. We observed a higher intensity and greater nuclear localization of p65 in an APOE genotype- and sex-dependent manner, with APOE4 microglia exhibiting high levels of total p65. Furthermore, female APOE4 cultures had higher total p65 levels than male microglia. We also assessed the levels of nuclear p65, and similar to total p65, sex-specific differences were observed in APOE4 cultures basally, as well as after LPS + IFNg treatment, with female APOE4 cultures exhibiting more p65 in their nuclei compared to male cultures. This highlights a potential inflammatory signaling pathway involved in the differential inflammatory profiles of APOE genotype- and sex-specific microglia. These findings add to the growing literature indicating the importance of including both the APOE genotype and sex when studying microglia-mediated inflammation in AD.

Furthermore, the APOE4 genotype is linked to a partial loss of function [96], and its inefficacy to suppress an inflammatory response [23] could be a plausible explanation for an increased inflammatory profile in the APOE4 genotype. APOE plays a vital role in transporting cholesterol [97], and previous studies have shown that the APOE4 genotype has a reduced capacity in this transport function [98]. Recent studies on APOE targeted replacement mice have reported altered lipid metabolism and impaired cholesterol trafficking in APOE4 microglia [99]. Since dysregulation in cholesterol/lipid homeostasis has been linked to inflammation [100,101,102], this may contribute to APOE genotype- and sex-specific inflammation. The APOE4 genotype has also been shown to disrupt autophagy-lysosomal processes [103]. Altered autophagic processes modulate immune functions and are linked to inflammation [104,105], which may also contribute to the higher inflammatory state of APOE4 microglia.

Although the current study was conducted on microglia from the whole brain, recent evidence has indicated that microglia possess regional transcriptional heterogeneity and region-specific functions [106,107]. Microglial genes from the cortex show upregulation in an immune-activated microglial phenotype, whereas the hippocampal microglial gene cluster is highly involved in bioenergetic maintenance [106]. Peripheral endotoxin-induced inflammation has previously been shown to alter the gene expression of cytokines and immune cell markers in a brain region-specific manner, with microglia also demonstrating region specificity in the inflammatory profile [108,109]. A recent comprehensive study showed APOE4-associated differential regional vulnerability by utilizing transcriptomics analysis [110]. Since the frontal cortex and hippocampus are the most vulnerable regions for the developing AD pathology, investigating the region-specific activation of microglia could provide mechanistic insights. Further to this, we used neonatal mouse pups to generate primary microglial cultures. Evidence from the characterization of inflammation for neonatal and adult primary microglia shows a similar pattern of response to TLR agonists, including LPS. However, the magnitude of response differs, with adult primary microglia demonstrating a more robust response, suggesting that the data we report in neonatal microglia may be underestimating the effects observed in adults [111]. Future studies will focus on profiling inflammation from adult/aged animals, thus defining the in vitro modulation of genotype and sex by age differences since aging influences the glial phenotype and molecular profile [112,113]. In a final point to note, our study was performed in primary microglial cultures. Although studies on primary microglia can aid immensely in understanding the underlying mechanisms involved in cell-specific responses, culturing primary cells disrupts the normal cellular interactions and may result in slight alterations of the transcriptional profile and functional activity [114]. Thus, studies conducted ex vivo will be needed for confirmation in more complex systems.

Neuroinflammation is a key event occurring in several neurodegenerative diseases. Previous studies have reported elevated blood and brain endotoxin levels in AD [115,116]. LPS has been widely used as an inflammatory stimulus for both in vitro and in vivo studies, including AD. Systemic LPS challenge, as well as injections in different regions of the CNS, have traditionally been used to mimic neuroinflammation in AD [117]. Studies in human AD brains indicate a seven-fold greater accumulation of LPS in the neocortical neurons compared to age-matched control patients [118]; moreover, LPS is associated with amyloid plaques in the brain [119]. Recent studies have shown that rats exposed to functional bacterial amyloid protein curli, made by *Escherichia coli*, had enhanced microgliosis [120] and is structurally similar to human amyloid-β and serum amyloid A [121]. Therefore while there is a rationale for evaluating the effects of LPS and LPS + IFNg, it is only one step in comprehending the sex-specific inflammatory profile of primary microglia from TR-APOE mice. Further studies are required to evaluate the effects of curli and amyloid-beta species, to explore neuroinflammation in AD.

## 4. Materials and Methods

### 4.1. Animals

TR-APOE breeder mice with a C57BL/6 background, with the mouse APOE gene replaced by a homozygous human APOE3 or APOE4 gene [50], were purchased from Taconic and bred in-house. Breeders were given food and water ad libitum and housed with a 12-h light-dark cycle. Primary microglial cultures were obtained from postnatal TR-APOE3 and TR-APOE4 pups aged 0–1 day. The biological sex of the mouse pups was determined by measuring the anogenital distance and the presence or absence of visible pigmentation in the anogenital region [122]. For mixed-sex cultures, approximately 50% male and female pups were used for each isolation. All procedures were conducted in accordance with the National Institutes of Health’s Guide for Care and Use of Laboratory Animals, the ARRIVE guidelines, and approved by the Institutional Animal Care and Use Committee of Florida International University.

### 4.2. Reagents

All cell culture reagents were purchased from Invitrogen (Waltham, MA, USA), with the exception of charcoal-stripped fetal bovine serum (cat. no. F6765) purchased from Sigma-Aldrich (St. Louis, MO, USA). Lipopolysaccharide (LPS; *Escherichia coli* O127:B8, cat. no. L4516) was purchased from Sigma-Aldrich, and interferon-gamma (IFNg; cat. no. 485-MI-100) was purchased from R & D Systems (Minneapolis, MN, USA).

### 4.3. Primary Mouse Microglia (PMG) Isolation

Mixed-sex and sex-specific PMG cultures were isolated from postnatal 0–1 day old, targeted replacement APOE3 or APOE4 mouse pups as previously described [123,124]. Briefly, whole brains were collected, washed in 1× phosphate-buffered saline, trypsinized, thoroughly triturated, and passed through a 70 μm strainer. This mixed glial culture was maintained in flasks for 15 days in Dulbecco’s modified Eagle’s medium/F12 (DMEM/F12) supplemented with 10% charcoal-stripped heat-inactivated serum (HI-FBS), 2 mM L-glutamine, 1 mM sodium pyruvate, 100 µM non-essential amino acids, 1% penicillin/streptomycin, and 2.5 μg/mL Plasmocin™. Charcoal-stripped fetal bovine serum was used to ensure no exogenous source of hormones interfered with the sex-specific results. Full media changes were done every 5–6 days with this growth medium. After 15 days, PMG were separated from other cell types by a CD11b immunomagnetic positive selection kit (cat. no. 18970, STEMCELL Technologies, Vancouver, BC, Canada) and cultured in the growth medium.

Following isolation, the cell purity of PMG was assessed by imaging 3–4 random fields per well (3–4 wells) in a 96-well plate using the 10× objective of a Keyence BZ-X810 imaging instrument. Cells positive for Iba1 (microglia) or GFAP (astrocytes) were counted against the total number of cells (positive for DAPI) per field using BZ-X800 Analyzer software, and the average was determined the percentage of purity. Cultures of greater than 98% purity were consistently obtained. Before the treatments, sex specificity of the cells was validated by examining the presence or absence of the sex-determining region-Y (*Sry*) gene by qPCR. Each isolation was derived from a separate litter obtained from different parents.

### 4.4. Treatments

For cytokine and media nitrite measurement, PMG were seeded at a density of 140,000 cells/well in 96-well plates. For qPCR experiments, 500,000 cells/well were seeded in 24-well plates. For each isolation (N), cells were seeded in two separate plates (*n*), with two to three replicates for each treatment group per plate. Treatments were performed in growth media supplemented with 2% charcoal-stripped HI-FBS. PMG were treated with the vehicle control (growth media containing 2% charcoal-stripped HI-FBS), LPS (10 ng/mL), or a combination of LPS (10 ng/mL) + IFNg (10 ng/mL).

### 4.5. Conditioned Media Collection

Primary microglia were treated with the vehicle control, LPS, or a combination of LPS + IFNg for 24 h, during which cytokines were allowed to accumulate. After 24 h, the media was briefly spun down, then the supernatant was collected and stored in aliquots at −80 °C until further use.

### 4.6. Measurement of Secreted Cytokines

Secreted cytokines were measured in the conditioned media using a custom U-PLEX mouse panel for the following analytes, IL1b, IL6, TNFa, IFNg, MCP1, MIP-1a, IL-4, and IL10, on the Mesoscale Discovery (MSD) platform following the manufacturer’s instructions. Plates were read on MSD QuickPlex SQ 120 and analyzed using the Discovery Workbench software.

### 4.7. Measurement of Media Nitrite

Nitric oxide is a signaling molecule that plays a key role in the pathogenesis of neuroinflammation and neurodegeneration [125]. Media nitrite is one of the two stable products of nitric oxide breakdown. Media nitrite was quantified using the Griess reagent system (Promega) as previously described [126]. Following 24 h of treatment, the media were collected and the nitrite levels were measured at an absorbance of 535 nm along a nitrite standard reference curve, as per the manufacturer’s protocol.

### 4.8. Quantitative RT-qPCR

PMG were treated with LPS in the presence or absence of IFNg for 6 h. RNA was isolated using TRIzol Reagent following the manufacturer’s instructions. cDNA was synthesized using All-in-One cDNA Synthesis Supermix (Bio-Rad, Hercules, CA, USA). Subsequent PCR reactions were performed using Itaq^TM^ Universal SYBR green (Bio-Rad, Hercules, CA, USA) following the manufacturer’s instructions and as previously described [127]. A list of genes and primer sequences used in qPCR reactions is presented in Appendix A. Gene expression was normalized to Rpl13a using the 2^-ΔΔCT^ method with the threshold cycle (CT) value for the housekeeping gene Rpl13a and the respective gene of interest [128].

### 4.9. Immunocytochemistry

PMG were seeded in 96-well plates at 140,000 cells/well, fixed with 4% paraformaldehyde (PFA) for 15 min, and washed with PBS, then immunocytochemistry was performed as previously described [126]. To assess the purity of the cultures, cells were permeabilized with PBS + 0.5% Triton X-100 and blocked for 1 h in 2% bovine serum albumin (BSA) solution at room temperature, followed by overnight incubation in 2% BSA with the following primary antibodies, GFAP (1:1000, cat. no. ab4674, Abcam, Cambridge, UK) and Iba1 (1:1000, cat. no. 019-19741, Wako Chemicals, Richmond, VA, Canada), at 4 °C. The following day, cells were washed with PBS and incubated with Alexa Fluor^®^ secondary antibodies from appropriate species (1:1000, ThermoFisher Scientific, Waltham, MA, USA) for 1 h at room temperature. After washing with PBS, cells were incubated with 10 μg/mL DAPI for 15 min, washed again, and visualized under the microscope. The staining specificity was confirmed by omitting the primary or secondary antibody. Cells were imaged using the Keyence BZ-X810 microscope at 10× magnification (N.A. 0.45, PlanApo lens).

For p65 staining, cells were fixed with 4% PFA, washed with 1× PBS, and permeabilized and blocked with 0.5% Triton X-100 and 0.005% Tween-20 in 2% BSA for 1 h at room temperature. PMG were then incubated overnight in a cocktail of p65 (1:400, cat. no. PA1-186, Invitrogen, Waltham, MA, USA) and IBA1 (1:1000) primary antibodies at 4 °C. The following day, cells were washed and incubated with the appropriate Alexa Fluor^®^ secondary antibodies (1:1000, ThermoFisher Scientific, Waltham, MA, USA) for 1 h at room temperature, stained with DAPI, and visualized under the microscope. Representative images were taken at 60× (N.A. 0.70, SuperFluo lens). For quantification of p65 levels, three images/well were taken at 10× magnification from triplicates of the treatment for each isolation. The total p65 was quantified in individual cells using the BZ-X800 Analyzer Hybrid Cell Count Software (Keyence). Parameters were set to a threshold and applied automatically to all images. Fluorescence intensity was normalized to the number of cells per field and calculated utilizing brightness and cell counts within the software. For nuclear p65 quantification, the single color extraction module of the BZ-X800 Analyzer Hybrid Cell Count Software was used. Nuclei of cells stained with DAPI were outlined as the target area. The integrated density of the p65 stain was measured in the nuclei and normalized to the total number of cells per field of the image taken.

### 4.10. Statistics

Statistical analysis was performed using Prism 7.04 (GraphPad Software, San Diego, CA, USA). Basal differences were analyzed in mixed-sex and sex-specific cultures by an unpaired Student’s t-test and two-way analysis of variance (ANOVA), respectively. For comparisons, including LPS and LPS + IFNg treated groups, mixed-sex cultures were analyzed using two-way ANOVA, whereas sex-specific cultures were analyzed by three-way ANOVA. Tukey’s post-hoc test was conducted to compare multiple groups following ANOVA. Statistical significance was considered as *p* < 0.05. Heat maps were generated using Displayr software (Pyrmont, Australia). Graphical abstract was created using BioRender.com.

## 5. Conclusions

Collectively, these data indicate that the APOE4 genotype and female sex together contribute to a greater inflammatory response in microglia isolated from targeted replacement APOE mice. Currently, there is a lack of comprehensive assessments of sex differences in microglia stratified by the APOE genotype. While effects of inflammation have been studied independently in a sex-specific or an APOE isoform-dependent manner, very few studies have investigated how the APOE genotype and sex together modulate the response to inflammation [88]. These findings provide a compelling rationale for the inclusion of both male and female sexes when evaluating the role of inflammation in AD. To our knowledge, the results presented in this study are the first data from TR-APOE mice to show significant sex-specific effects after an inflammatory stimulus in cultured microglia. These data are consistent with clinical data, and sex-specific microglia may provide a platform for exploring mechanisms of genotype and sex differences in AD, particularly as they relate to neuroinflammation and neurodegeneration.

## Figures and Tables

**Figure 1 ijms-23-09829-f001:**
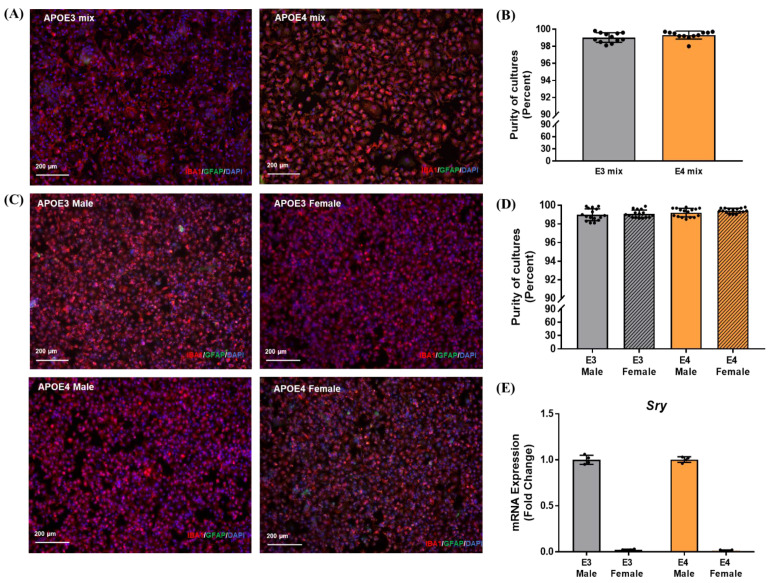
Purity of mixed-sex and sex-specific microglial cultures: (**A**) Representative images for purity of mixed-sex microglial cultures as determined by staining for IBA1 (microglia), GFAP (astrocyte), and DAPI (nucleus); (**B**) purity of mixed-sex microglial cultures was greater than 98%; (**C**) representative images for purity of sex-specific microglial cultures; (**D**) purity greater than 98% was consistently obtained for all sex-specific microglial cultures. Each data point in (**B**,**D**) denotes the average percent of microglia from three to four images per well. (**E**) Sex-specificity of cultures was validated by checking for the presence or absence of sex-determining region-Y (*Sry*) by qPCR. Each data point represents the average mRNA expression from one isolation. Data are represented as the mean ± standard deviation.

**Figure 2 ijms-23-09829-f002:**
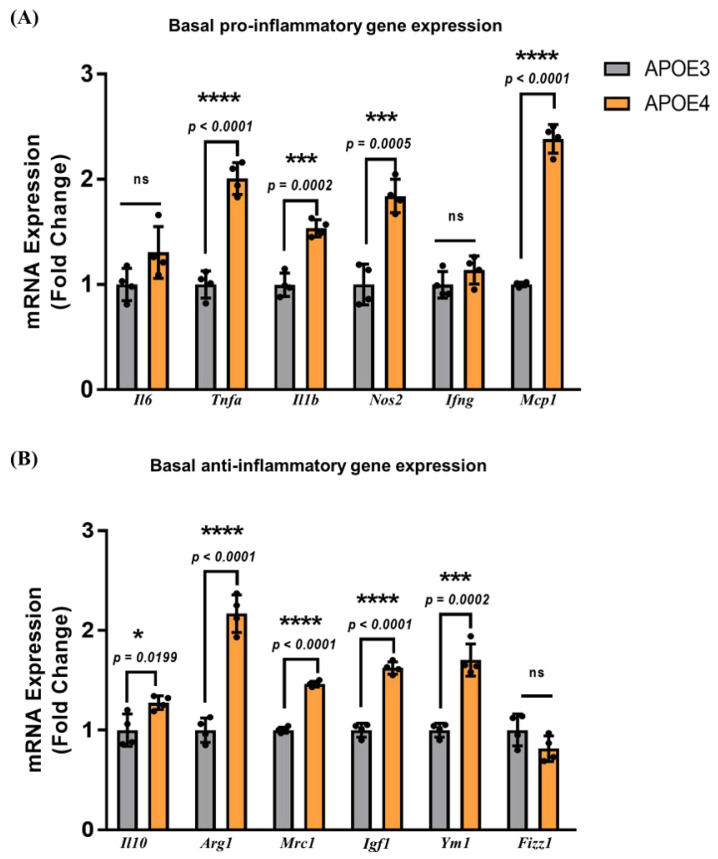
Basal pro− and anti−inflammatory gene expression in mixed−sex microglia is APOE genotype−dependent: (**A**) basal pro−inflammatory and (**B**) anti−inflammatory gene expression was evaluated in non−stimulated mixed−sex microglia from APOE3 and APOE4 genotypes. Gene expression between genotypes was analyzed using an unpaired Student’s *t*−test (N = 3 isolations, *n* = 6 plates). Data are represented as the mean ± standard deviation. *, ***, and **** indicate statistical significance for the comparison and denote *p* < 0.05, 0.001, and 0.0001, respectively. ns = not statistically significant (*p* > 0.05).

**Figure 3 ijms-23-09829-f003:**
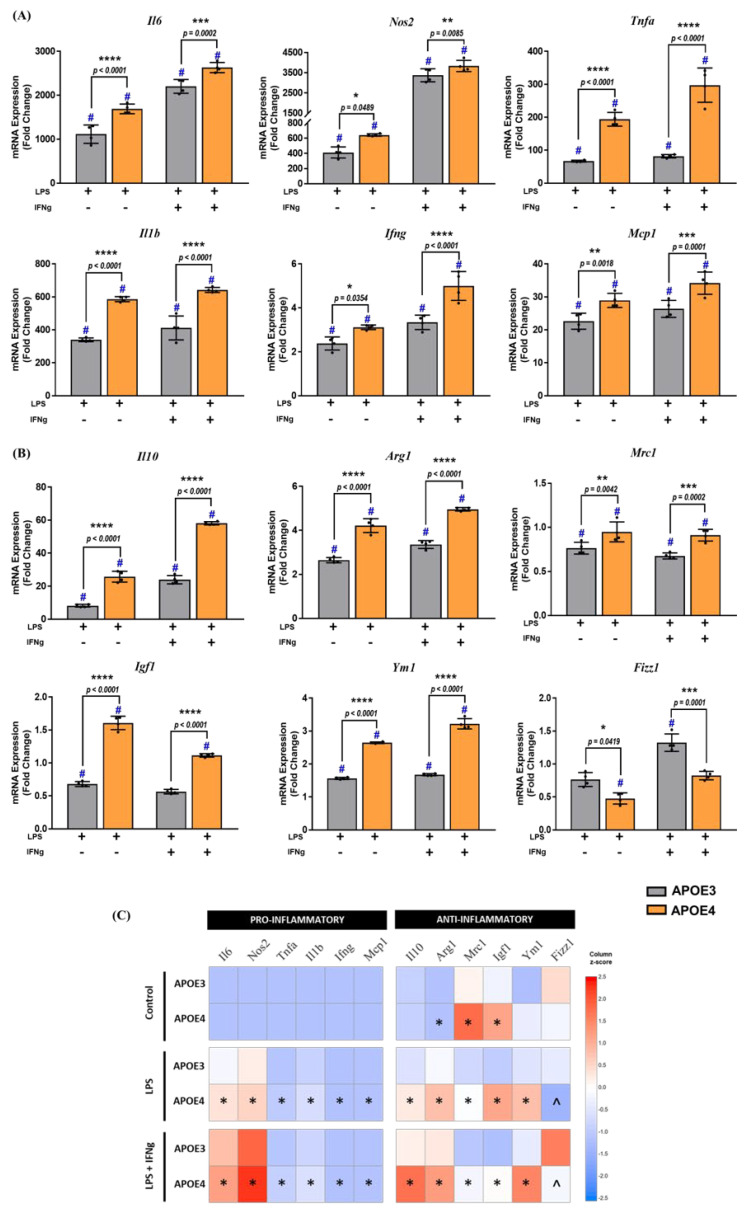
Comparison of gene expression between APOE3 and APOE4 genotypes: (**A**) pro− and (**B**) anti−inflammatory gene expression was evaluated in mixed−sex microglia from APOE3 and APOE4 genotypes following treatment with LPS (10 ng/mL) and LPS (10 ng/mL) + IFNg (10 ng/mL) for 6 h. Gene expression between genotypes was assessed by two−way ANOVA followed by Tukey’s multiple comparisons test (N = 3 isolations, *n* = 6 plates). Each data point represents the average mRNA expression from one isolation. Data are represented as the mean ± standard deviation. # denotes a significant increase compared to the respective genotype control (data not included in the graphs, *p* < 0.05). *, **, ***, and **** indicate statistical significance for the comparison and denote *p* < 0.05, 0.01, 0.001, and 0.0001, respectively. (**C**) Heat map comparing the mean expression of pro− and anti−inflammatory genes in control, LPS, and LPS + IFNg treated APOE3 and APOE4 PMG. * and ^ denote a significant (*p* < 0.05) increase and decrease, respectively, in APOE4 gene expression compared to APOE3.

**Figure 4 ijms-23-09829-f004:**
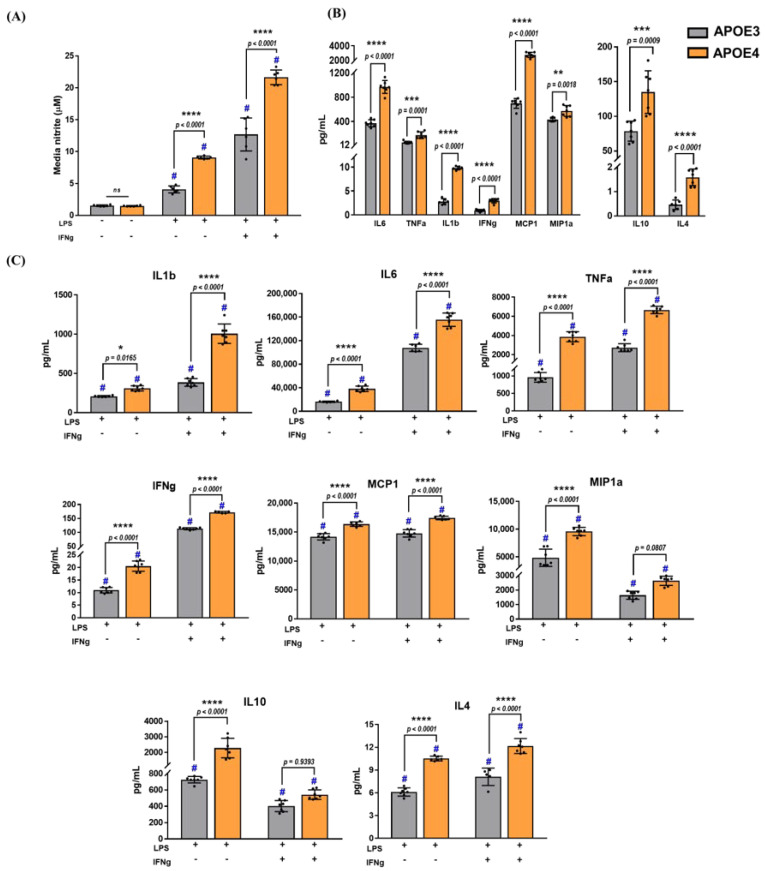
APOE4 genotype exacerbates release of pro− and anti−inflammatory cytokines: cytokine levels were measured following treatment with LPS or LPS + IFNg for 24 h. (**A**) Average levels of nitrite (±standard deviation) accumulated in the media were measured by GRIESS assay. Using two−way ANOVA, significant treatment and genotype effects were observed. # denotes a significant increase compared to the respective genotype control (*p* < 0.05). (**B**) Basal levels of secreted cytokines were measured in non−stimulated microglia after 24 h by an MSD multiplex assay. Comparisons between genotypes were made using an unpaired Student’s *t*−test. (**C**) Cytokine levels in the media were measured after treatment with LPS or a combination of LPS and IFNg for 24 h. Two−way ANOVA was used to assess cytokine release in mixed−sex microglia from APOE3 and APOE4 genotypes (N = 3−4 isolations, *n* = 6−8 plates). Data are represented as the mean ±standard deviation, and each data point denotes the average of each experimental plate. # denotes a significant increase compared to the respective genotype control (data not included in the graphs, *p* < 0.05). *, **, ***, and **** indicate statistical significance for the comparison and denote *p* < 0.05, 0.01, 0.001, and 0.0001, respectively. ns = not statistically significant (*p* >0.05).

**Figure 5 ijms-23-09829-f005:**
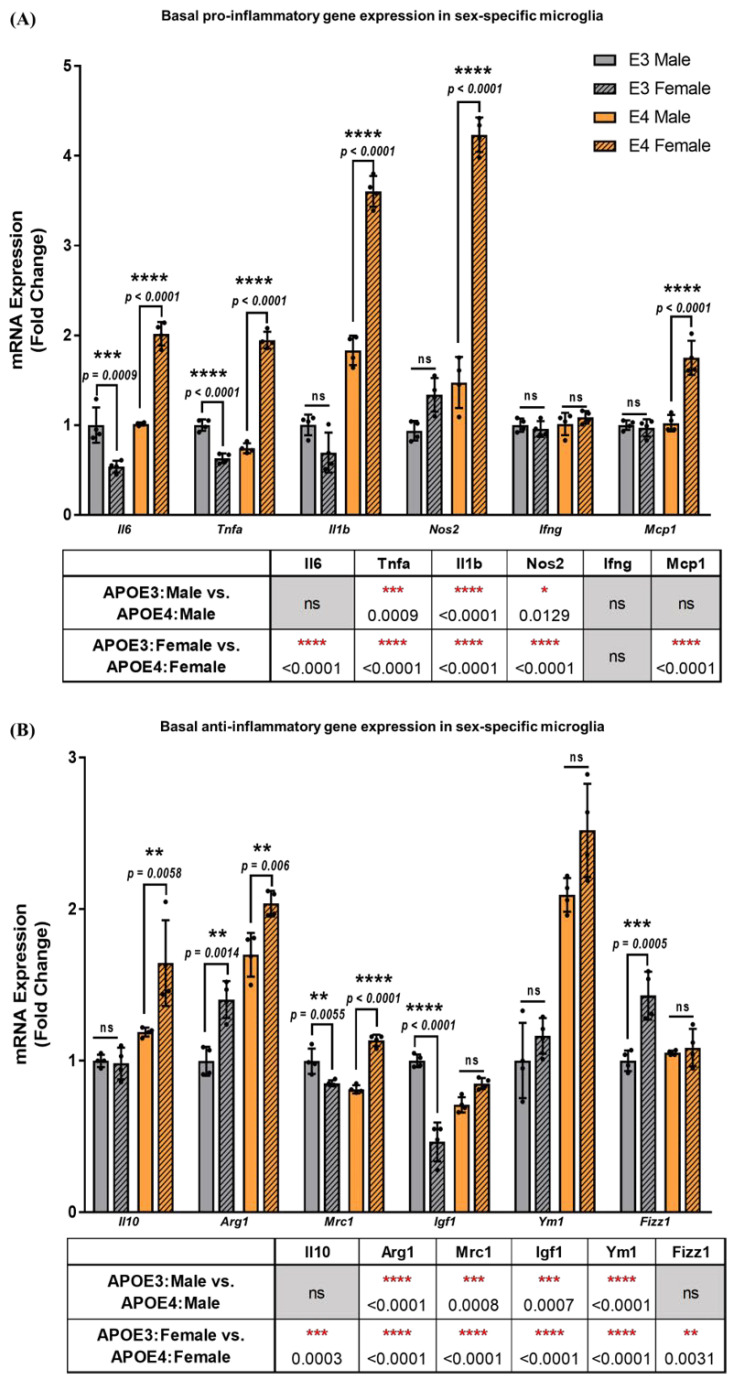
Basal pro− and anti−inflammatory gene expression in sex−specific microglia is dependent on both the APOE genotype and sex: (**A**) basal pro−inflammatory and (**B**) anti−inflammatory gene expression was evaluated in non−stimulated sex−specific microglia from APOE3 and APOE4 genotypes. Gene expression between genotypes and sexes was analyzed using two−way ANOVA (N = 3 isolations, *n* = 6 plates). Data are represented as the mean ± standard deviation. *, **, ***, and **** indicate statistical significance for the comparison and denote *p* < 0.05, 0.01, 0.001, and 0.0001, respectively. ns = not statistically significant (*p* > 0.05).

**Figure 6 ijms-23-09829-f006:**
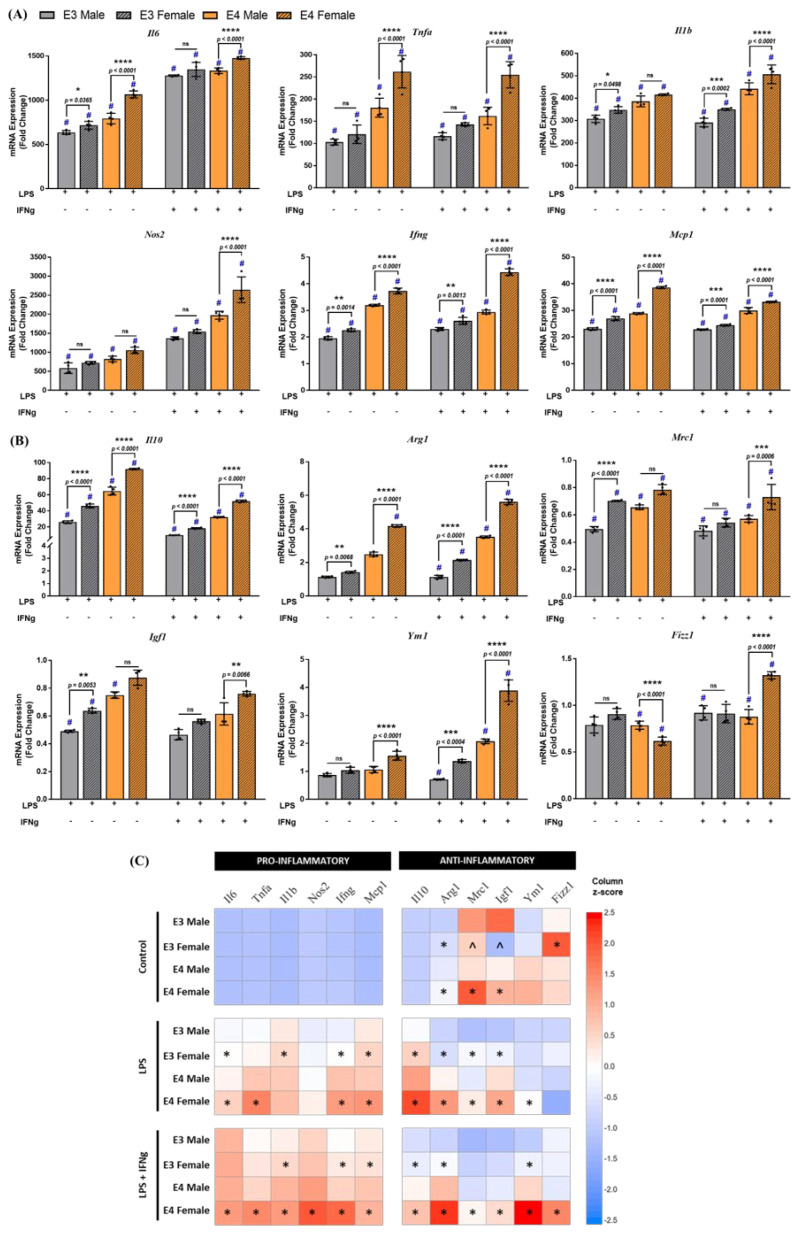
Gene expression in sex-specific microglia is dependent on both the APOE genotype and sex: (**A**) pro− and (**B**) anti−inflammatory gene expression was evaluated in sex−specific microglia from APOE3 and APOE4 genotypes following LPS or LPS + IFNg treatment for 6 h. Gene expression between genotypes and sexes was analyzed using a three−way ANOVA (N = 3 isolations, *n* = 6 plates) followed by Tukey’s multiple comparisons test. Each data point represents the average mRNA expression from one isolation. Data are represented as the mean ± standard deviation. # denotes a significant increase compared to the respective genotype control (data not included in the graphs, *p* < 0.05). *, **, ***, and **** indicate statistical significance for the comparison and denote *p* < 0.05, 0.01, 0.001, and 0.0001, respectively. ns = not statistically significant (*p* >0.05). (**C**) Heat map comparing the mean expression of pro− and anti−inflammatory genes in sex−specific APOE3 and APOE4 PMG treated with LPS and LPS + IFNg. * and ^ denote a significant (*p* < 0.05) increase and decrease, respectively, in fold changes of gene expression compared to a genotype−matched male PMG.

**Figure 7 ijms-23-09829-f007:**
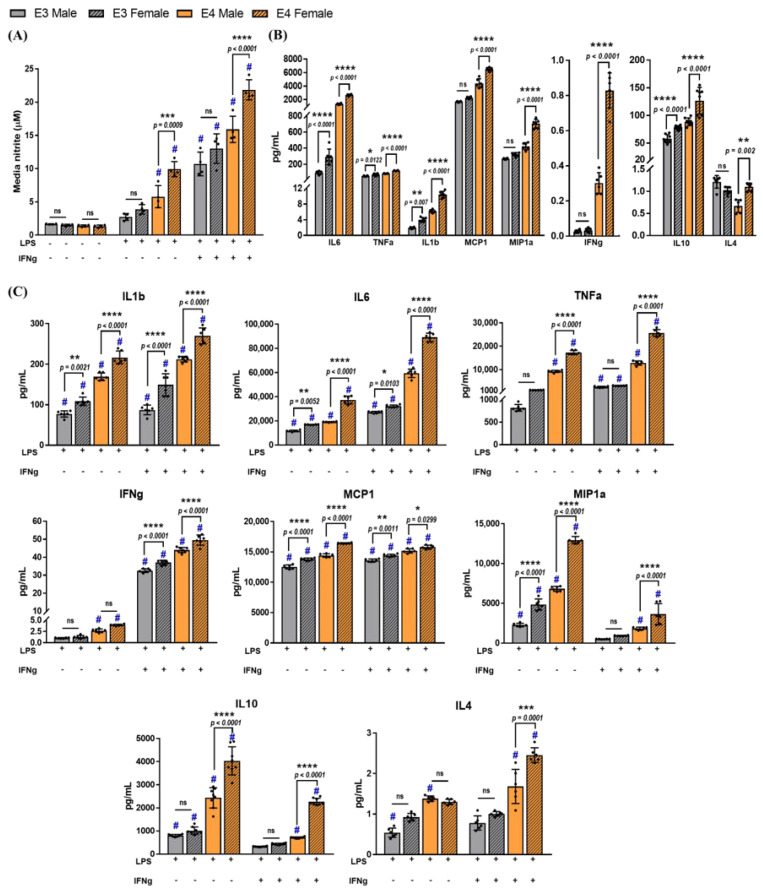
Increased secretion of inflammatory cytokines is dependent on both the APOE genotype and sex: (**A**) average nitrite levels accumulated in the media were measured by GRIESS assay. Three−way ANOVA followed by Tukey’s multiple comparisons test revealed a significant treatment−by−genotype−by−sex effect. # denotes a significant increase compared to the respective genotype control (*p* < 0.05). (**B**) Basal levels of secreted cytokines were measured in non−stimulated microglia after 24 h by an MSD multiplex assay. Comparisons between sex and genotype were made using two-way ANOVA followed by Tukey’s multiple comparisons test (N = 3 isolations, *n* = 6 plates). (**C**) Secretion of cytokines was measured post−treatment with LPS or LPS + IFNg by a three−way ANOVA followed by Tukey’s multiple comparisons test. Data are represented as the mean ± standard deviation. # denotes a significant increase compared to the respective control (data not included in the graphs, *p* < 0.05). *, **, ***, and **** indicate statistical significance for the comparison and denote *p* < 0.05, 0.01, 0.001, and 0.0001, respectively. ns = not statistically significant (*p* > 0.05).

**Figure 8 ijms-23-09829-f008:**
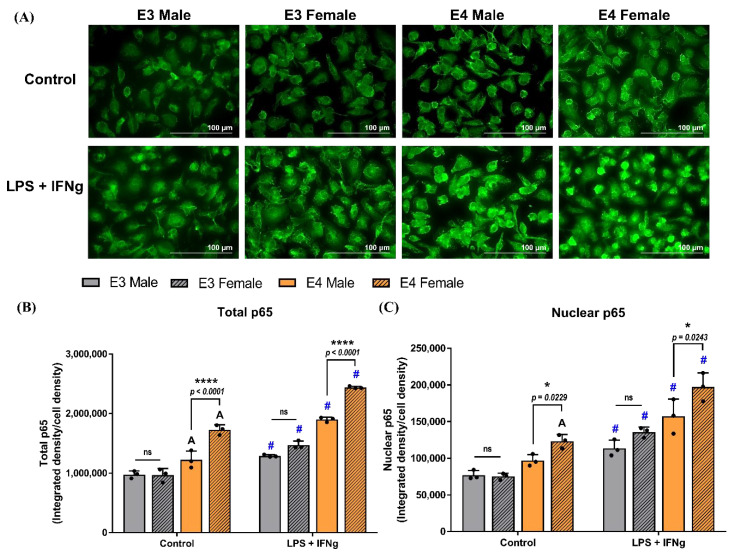
APOE genotype and sex-dependent inflammation are mediated through p65 activation: (**A**) representative images for immunocytochemistry of p65 in control and LPS + IFNg treated APOE3 and APOE4, male and female PMG (scale bar = 100 μm). (**B**) Quantification of total p65 and (**C**) quantification of nuclear p65. N = 3 isolations. Each data point denotes the average p65 integrated density/cell density for an individual isolation. A denotes a significant increase compared to the E3 male control (*p* < 0.05). # denotes a significant increase compared to the respective genotype and sex control (*p* < 0.05). Data are represented as the mean ± standard deviation. * and **** indicate statistical significance for the comparison and denote *p* < 0.05 and 0.0001, respectively. ns = not statistically significant (*p* > 0.05).

## Data Availability

All data generated or analyzed during this study are included in this published article. Further information/data are available from the corresponding author on reasonable request.

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
