# Peer review of "Sex and APOE Genotype Alter the Basal and Induced Inflammatory States of Primary Microglia from APOE Targeted Replacement Mice"

_ijms, 2022, doi:10.3390/ijms23179829_

Round 1
Reviewer 1 Report
The current manuscript is very well written and is very focused upon the inflammatory signaling pathway associated with APOe3 and 4. The methods is mostly cell culture methodology and reduces the enthusiasm. Further APOe signaling is associated with cholesterol transport. This would include signaling molecules from APOe4/3 molecules being excreted. Second potential influence on autophagy or ability to reduce hallmark pathologies associated with Alzheimer's disease. In addition inflammatory cells including macrophages, T-cells, and monocytes are also influenced by APOe signaling for cholesterol export. It is suggested that in the discussion a weakness of the current study is that alternative inflammatory cells can also be involved in inflammation. Lastly, there needs to have some inflammatory signaling pathway associated which shows similarities or differences between signaling pathways in males vs females. Lastly, findings from the current report should be taken with caution, because cells are isolated from the host, and thus lose the response of what occurs in vivo. In addition what if these cells were isolated from an APOe4x5xFAD animal or a tau animal would there be more differences to understand the impact of APO4/3 on inflammation in Alzheiemr's disease.
Author Response
Reviewer1:
The current manuscript is very well written and is very focused upon the inflammatory signaling pathway associated with APOe3 and 4. The methods is mostly cell culture methodology and reduces the enthusiasm. Further APOe signaling is associated with cholesterol transport. This would include signaling molecules from APOe4/3 molecules being excreted. Second potential influence on autophagy or ability to reduce hallmark pathologies associated with Alzheimer's disease. In addition inflammatory cells including macrophages, T-cells, and monocytes are also influenced by APOe signaling for cholesterol export. It is suggested that in the discussion a weakness of the current study is that alternative inflammatory cells can also be involved in inflammation. Lastly, there needs to have some inflammatory signaling pathway associated which shows similarities or differences between signaling pathways in males vs females. Lastly, findings from the current report should be taken with caution, because cells are isolated from the host, and thus lose the response of what occurs in vivo. In addition what if these cells were isolated from an APOe4x5xFAD animal or a tau animal would there be more differences to understand the impact of APO4/3 on inflammation in Alzheiemr's disease.
We appreciate the comments and suggestions by the reviewer.
- We agree that there may be several potential pathways attributing to the differences in the inflammatory profiles of APOE and sex-specific microglia, including variation in Apoe protein levels, cholesterol trafficking, and autophagy. We have expanded the discussion to include this.
- We have added immunofluorescence experiments assessing the activation of the p65 subunit of the transcription factor NF-κB. The original manuscript highlighted inflammatory endpoints but lacked mechanistic insight into the differential cytokine levels observed. Thus, quantification of total and nuclear p65 provides a potential inflammatory signaling pathway linking the APOE genotype and sex-specific differential inflammatory profiles.
- We have revised the discussion per the reviewer's suggestions and addressed the caveats of this study (line 569).
- The authors agree and think that investigating APOE genotype and sex differences in the APOE x 5xFAD or tau mouse model would be good experiments for the future. However, since the current study focuses on the sporadic nature of Alzheimer’s disease and the 5xFAD mouse model harbors familial AD mutations, we did not use these models here.
Reviewer 2 Report
This is a comprehensive analysis to examine how sex and ApoE genotype at cellular level can generate different inflammatory responses.
In figure 1, it will be good for authors to include the protein markers used to assess the purity of isolated culture in the figure legend. Currently, they are mentioned in methods.
Should conclusions come after discussion and not after methods?
Author Response
Reviewer2:
This is a comprehensive analysis to examine how sex and ApoE genotype at cellular level can generate different inflammatory responses.
In figure 1, it will be good for authors to include the protein markers used to assess the purity of isolated culture in the figure legend. Currently, they are mentioned in methods.
Thank you for bringing this to our attention. We have added the names of the protein markers used to assess purity in the figure legend (line 120).
Should conclusions come after discussion and not after methods?
Thank you for raising this point. We followed the Microsoft Word template provided online by IJMS https://www.mdpi.com/journal/ijms/instructions . Please advise if this is not the correct sequence.